

# Phylogenetic relationships of Neogene hamsters (Mammalia, Rodentia, Cricetinae) revealed under Bayesian inference and maximum parsimony

Moritz Dirnberger[1], Pablo Peláez-Campomanes[2] and
Raquel López-Antoñanzas[1]

[1] ISEM, University of Montpellier, CNRS, IRD, Montpellier, France
[2] Departamento de Paleobiología, Museo Nacional de Ciencias Naturales-CSIC, Madrid, Spain

## ABSTRACT

There is an ongoing debate about the internal systematics of today's group of hamsters (Cricetinae), following new insights that are gained based on molecular data. Regarding the closely related fossil cricetids, however, most studies deal with only a limited number of genera and statements about their possible relationships are rare. In this study, 41 fossil species from the Late Miocene to the Pliocene, belonging to seven extinct cricetine genera, *Collimys*, *Rotundomys*, *Neocricetodon*, *Pseudocricetus*, *Cricetulodon*, *Apocricetus* and *Hattomys* are analysed in a phylogenetic framework using traditional maximum parsimony and Bayesian inference approaches. Following thorough model testing, a relaxed-clock Bayesian inference analysis is performed under tip-dating to estimate divergence times simultaneously. Furthermore, so-called 'rogue' taxa are identified and excluded from the final trees to improve the informative value of the shown relationships. Based on these resulting trees, the fit of the topologies to the stratigraphy is assessed and the ancestral states of the characters are reconstructed under a parsimonious approach and stochastic character mapping. The overall topologies resulting from Bayesian and parsimonious approaches are largely congruent to each other and confirm the monophyly of most of the genera. Additionally, synapomorphies can be identified for each of these genera based on the ancestral state reconstructions. Only *Cricetulodon* turns out to be paraphyletic, while '*Cricetulodon*' *complicidens* is a member of *Neocricetodon*. Lastly, this work makes a contribution to a debate that went on for decades, as the genus *Kowalskia* can be confirmed as junior synonym of *Neocricetodon*.

## INTRODUCTION

Cricetidae, with more than 700 living species, is the second most speciose family inside Muroidea. According to molecular studies (*Musser & Carleton, 2005*; *Neumann et al., 2006*; *Steppan & Schenk, 2017*), it comprises the following subfamilies: the new World rats and mice (Sigmodontinae, Neotominae and Tylomyinae), the group of voles, musk rats

Corresponding author
Moritz Dirnberger,
dirnberger.moritz@umontpellier.fr

and lemmings (Arvicolinae), and the Old World hamsters (Cricetinae). Cricetinae (commonly known as hamsters), is a group of mouse to rat-sized rodents with cheek pouches and short tails, which comprises nowadays 18 species distributed in seven genera. Today, they live in the Palearctic realm, mostly in steppe and grassland habitats but also in desert areas and urban environments (*Pardiñas et al., 2017*). Over the history, hamsters have been considered either as a tribe (*e.g.*, *Simpson, 1945*) or a subfamily (*e.g.*, *Mein & Freudenthal, 1971*). The different taxonomic ranks attributed to this group have resulted from the lack of consensus concerning the taxonomic rank of Cricetidae, which has been classified as a subfamily inside Muridae with all its main clades treated as tribes instead of subfamilies (*McKenna & Bell, 1997*) or it has been considered as a family on its own (*Chaline, Mein & Petter, 1977*), which agrees with morphological and molecular reconstructions, with the exclusion of some genera from the group, however (*Steppan & Schenk, 2017*; *López-Antoñanzas et al., in press*).

Depending on the fossils attributed to Cricetinae, its temporal range varies from the Early Miocene (*e.g.*, *Mein & Freudenthal, 1971* who included *Democricetodon* within cricetines) or from the Middle Miocene (*e.g.*, *Fejfar et al., 2011* with *Collimys* as the earliest cricetine) until nowadays. In this work, we consider Democricetodontinae (including *Democricetodon* and *Copemys*, among other genera) to be stem Cricetidae among which we may find its potential ancestors (*Lindsay, 2008*; *López-Antoñanzas et al., in press*). Therefore, as a working hypothesis, we treat the cricetines as having a temporal range that spans from the Middle Miocene until today as considered by *Daxner-Höck (1972)* and *Fejfar et al. (2011)*.

This study does not include these stem cricetids but focuses on the earliest representatives of the subfamily Cricetinae: *Apocricetus Freudenthal, Mein & Martín Suárez, 1998*, *Collimys Daxner-Höck, 1972*, *Cricetulodon Hartenberger, 1965*, *Hattomys Freudenthal, 1985*, *Neocricetodon Schaub, 1934*, *Pseudocricetus Topachevsky & Skorik, 1992* and *Rotundomys Mein, 1965*. Its objective is to elucidate the phylogenetic relationships inside this group, for which these early forms represent the most important initial radiation.

Previous phylogenetic reconstructions merely focused on species belonging to one or two genera and were based on maximum parsimony solely (*Cuenca Bescós, 2003*; *López-Antoñanzas, Peláez-Campomanes & Álvarez-Sierra, 2014*; *Sinitsa & Delinschi, 2016*). Moreover, molecular phylogenetic studies dealing with extant Cricetinae incorporated fossil data solely to calibrate the nodes (*Steppan, Adkins & Anderson, 2004*; *Neumann et al., 2006*; *Steppan & Schenk, 2017*; *Lebedev et al., 2018*). However, additional approaches based on Bayesian methods have to be explored (see *López-Antoñanzas et al., 2022*) to shed light on the diversification processes of the studied groups and to be able to accurately estimate divergence times. Recent advances in this field include the so-called morphological clock, which refers to the rate of morphological changes through time. This rate together with the incorporation of fossils as tips, in order to calibrate the tree in a tip-dating approach, allows estimating divergence times, even in completely extinct clades (*Turner, Pritchard & Matzke, 2017*). The position of the fossil taxa on the tree is hereby simultaneously reconstructed. In this way, it is not necessary to rely on possibly wrong assumptions about

the position of fossil taxa, as is the case when applying the node-dating method (*Near, Meylan & Shaffer, 2005*; *Parham & Irmis, 2008*). Based on these ideas, more complex ways of modelling different aspects regarding a more accurate reconstruction of phylogenetic trees have been explored. This includes for example, relaxing the morphological clock rate (*Zhang, 2022*), incorporating a fossilized birth-death tree model (*Stadler, 2010*), or accounting for different taxon sampling strategies (*Höhna et al., 2011*).

In this study, we present the first reconstructed phylogeny based on dental morphological data, from a selected series of Late Miocene to Pliocene cricetine genera, mainly distributed in Europe, which exhibit high levels of species diversity. We compare the results obtained by applying different phylogenetic techniques, such as maximum parsimony and Bayesian inference approaches to propose the most robust phylogenetic hypothesis. Overall, the study contributes to a better understanding of the early evolution of the group and help to clear up previous systematic and taxonomic questions.

## MATERIALS AND METHODS

Upper molars are indicated with upper-case letters (M1, M2, M3), lower molars with lower-case letters (m1, m2, m3). The dental terminology used in this work is shown in Fig. 1.

### Taxon set

Included taxa depend on data availability and general completeness of the material. Within the seven extinct genera, *Apocricetus*, *Collimys*, *Cricetulodon*, *Hattomys*, *Neocricetodon* (including species assigned to *Kowalskia*), *Pseudocricetus* and *Rotundomys*, a total number of 41 species could be coded, which makes up around 77% of the total number of species within these genera (53), that were found in the literature. Additionally, two extant taxa were included as well, *Cricetus cricetus* (*Linnaeus, 1758*) and *Nothocricetulus migratorius* (*Pallas, 1773*). As outgroup, *Eucricetodon wangae Li, Meng & Wang, 2016* was added, as coded in *López-Antoñanzas & Peláez-Campomanes (2022)*. For additional information about the included taxa, *e.g.*, age interval, references, observed material, see Supplemental Material S1.

### Morphological characters and matrix construction

The matrix was constructed in Mesquite v. 3.81 (*Maddison & Maddison, 2023*). It is based on the morphological matrix from *López-Antoñanzas & Peláez-Campomanes (2022)*, and expanded here from 82 characters to 116 characters, introducing additional characters corresponding to the structures allowing to differentiate cricetine genera and species. Four characters concern the whole molar row, six refer to morphometrics, and the remaining 106 are related to the morphology of each dental element (M1: 37; M2: 11; M3: 16; m1: 23; m2: 9; m3: 10). In cases of intraspecific variability between different locations, only the condition found in the type locality has been considered. In case of variability in the type location, only the character state present in most of the specimens was taken into account, except for specimens, for which no clear majorities were seen. The morphological matrix is provided in Supplemental Material S2.

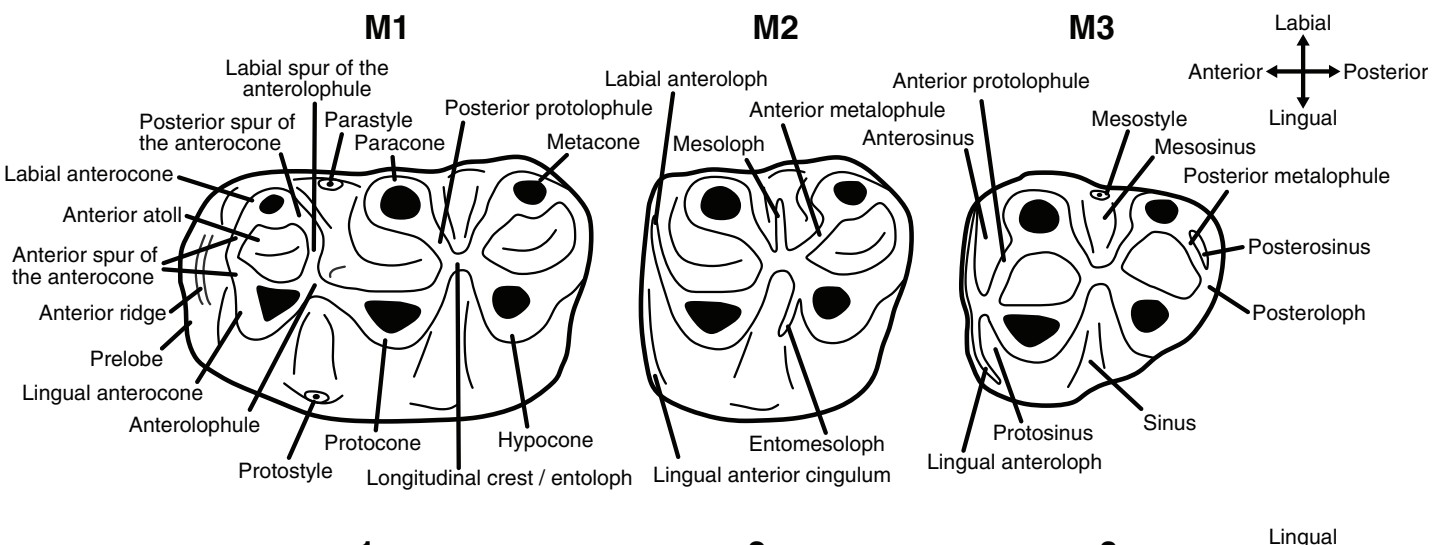

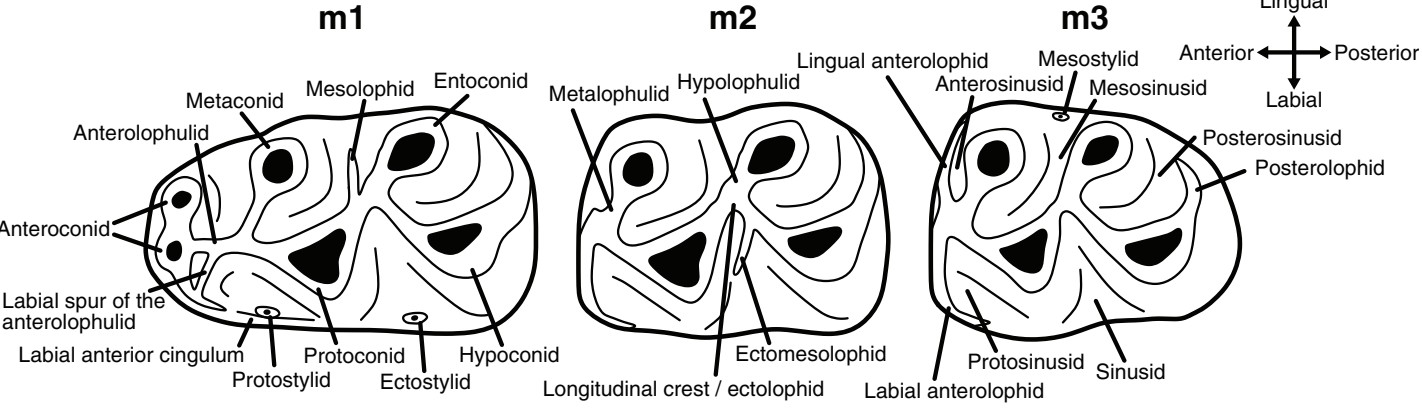

**Figure 1  Dental terminology used in this study.**

## Phylogenetic reconstructions

All final trees were annotated and visualized in R with the packages treeio, ggtree and deeptime (*Wang et al., 2020*; *Yu, 2022*; *Gearty, 2023*). For input and output files of the Bayesian inference and maximum parsimony analyses, see Supplemental Material S3.

**Maximum parsimony analyses.** All maximum parsimony analysis were run with TNT v. 1.6 (*Goloboff & Morales, 2023*) with all characters treated as unordered.

**Equal weights analysis.** The analysis under equal weights (MP-EW) was conducted with new technology algorithms using initial trees from 1,000 rounds of random addition sequence, with 100 iterations or rounds for sectorial search, ratchet, and tree fusing. The resulting 106 most parsimonious trees (256 steps, consistency index (CI): 0.414, retention index (RI): 0.715) were used to calculate a 50% majority consensus tree (258 steps, CI: 0.411, RI: 0.711). Clade support (given in %) was calculated based on 1,000 bootstrap (BS) replicates under the same parameters (*Felsenstein, 1985*).

**Implied weights analysis.** An additional analysis was run under the same options, as before but including implied weighting (MP-IW) (*Goloboff et al., 2008*). Following recent suggestions, a larger concavity index (*k*) of 12 was used (see *Goloboff, Torres & Arias,*

*2018*). The resulting two most parsimonious trees (257 steps, CI: 0.412, RI: 0.713) were used to calculate a consensus tree, (258 steps, CI: 0.411, RI: 0.711) with clade support based on 1,000 BS replicates (*Felsenstein, 1985*).

**Bayesian inference analyses.** Bayesian analyses were run with the parallel version of MrBayes v. 3.2.7a (*Altekar et al., 2004*; *Ronquist et al., 2012b*) using the Cyber-Infrastructure for Phylogenetic Research (CIPRES) Science Gateway version 3.3 (*Miller, Pfeiffer & Schwartz, 2010*).

**Non-clock analysis.** The Mkv model (*Lewis, 2001*) was used with among character rate heterogeneity modelled under a gamma distribution (*Yang, 1993*). All characters were treated as unordered. The analyses were run with four independent Metropolis-Coupled Markov chain Monte Carlo (MCMCMC) runs with six chains and 30,000,000 generations, sampling every 1,000 steps and a burn-in of 30%. Convergence and sufficient length of the runs were checked, using the R package Convenience v. 1.0.0 (*Fabreti & Höhna, 2022*). Based on the posterior tree sample a maximum clade compatibility (MCC) tree was calculated, as a consensus tree.

**Time-calibrated relaxed-clock analysis.** All settings of the non-clock analysis were adopted, except the number of MCMCMC generations, which was increased to 50,000,000. Time-calibrated relaxed-clock analyses were performed under a fossilized birth-death (FBD) tree prior (*Stadler, 2010*; *Zhang et al., 2016*). To model the way in which extant and extinct taxa are sampled in the construction of the tree, different strategies can be used (*Simões, Caldwell & Pierce, 2020*). To avoid problems when inferring speciation or extinction rates (*Höhna et al., 2011*), we have tested two of the three strategies, that are compatible with the FBD tree prior. The option 'diversity', that assumes a sampling strategy to maximize the diversity of extant taxa, was excluded, as our database only includes two extant species. Consequently, we have tested the two models that assume randomly sampled extant species. The first one, with sampled ancestors, SA-FBD ('random'), allows the fossil taxa to be tips or ancestors of other taxa, while in the second one, the so-called noSA-FBD ('fossiltip'), the fossil taxa have to be tips. The use of one or another can have an impact in the estimations of divergence times (*Gavryushkina et al., 2014*; *Simões, Caldwell & Pierce, 2020*). For the extant sampling probability, the number of included extant taxa (2) is divided by the total number of extant Cricetinae species (18 after *Musser & Carleton, 2005*).

In order to time-calibrate the tree, the tip-dating approach was used (*Ronquist et al., 2012a*; *Ronquist, Lartillot & Phillips, 2016*). Age ranges of the fossil taxa, resulting from age uncertainties of one or multiple locations of one taxon, were addressed by assigning uniform prior distributions to the tip calibrations, which can help to avoid erroneous divergence time estimations (*O'Reilly, Dos Reis & Donoghue, 2015*; *Barido-Sottani et al., 2019*). For the root age, an offset exponential distribution was set as a prior, with a minimum of 33 Ma (= minimal age of the oldest included fossil *Eucricetodon wangae*) and a mean of 41.2 Ma (following *López-Antoñanzas & Peláez-Campomanes, 2022*).

To give an informative prior to the base rate of the clock, the median tree length, calculated by a preceding non-clock analysis, was divided by the median of the root age prior (3.189768/37.1 = 0.085978) (following *Simões et al., 2018*, *2020*). This estimated rate

in natural log scale (= −2.45367) was used as the mean of a log-normal distribution with the exponent of the mean ($e^{0.085978}$ = 1.08978) as the standard deviation (following *Pyron, 2017*). To enforce proper rooting of the tree and facilitate reaching convergence, the ingroup was constrained to be monophyletic.

For relaxing the clock, two different models, compatible with the FBD prior are implemented in MrBayes v. 3.2.7a. The IGR (Independent gamma rate) model draws substitution rates from a gamma distribution, uncorrelated between branches, which allows more dramatic rate changes (more punctuated mode of evolution) (*Drummond et al., 2006*). The second model, TK02, samples from a lognormal distribution and is autocorrelated between branches, which represents a more gradual mode of evolution (*Thorne & Kishino, 2002*). Both models were used here, resulting in a total of four different models with all combinations of 'fossiltip' *vs*. 'random' and IGR *vs*. TK02. To choose the best fit model, stepping-stone sampling was done to estimate the marginal likelihoods (*Xie et al., 2011*). These can be used to calculate Bayes factors to compare the fit of two models to the data. For the stepping-stone sampling, the number of MCMCMC generations was increased by a factor of 10 to 500,000,000 as suggested by *Ronquist et al. (2020)*.

All analyses were checked for convergence by the R package Convenience v. 1.0.0 (*Fabreti & Höhna, 2022*), as mentioned for the non-clock analysis.

**Rogue taxon identification and tree set pruning.** To improve posterior probabilities of the resulting trees, so-called rogue ('wildcard') taxa, were identified. These taxa are characterised by an unstable position in the tree, as they are resolved in varying clades in the trees of a tree set, *e.g.*, the posterior tree sample of a Bayesian inference analysis. This leads to decreased posterior probabilities or even less resolved consensus trees. Equally, in the case of maximum parsimony analyses, they can affect the consensus tree calculated by several most parsimonious trees or the support values, given by a set of bootstrap trees. Deletion of the rogue taxa from the tree sets before calculating the consensus trees, can therefore lead to better resolved and supported trees, while they are still incorporated in the actual reconstruction. The deletion of the rogue taxa from the taxon set followed by a re-run of the analysis (*e.g.*, in *Aberer & Stamatakis, 2011*; *Simões, Caldwell & Pierce, 2020*) is seen critically by some authors as it means disregarding available and potentially important information (as discussed in *Goloboff & Szumik, 2015*).

In this study, the posterior tree samples of both clock trees were used to identify rogue taxa utilising the R package Rogue v. 2.1.6 (*Smith, 2022, 2023*). An additional examination of the 106 most parsimonious trees of the equal weighting maximum parsimony analysis, using the web interface of RogueNaRok (*Aberer, Krompass & Stamatakis, 2013*), did not result in any identified rogue taxa. For the R-code used to identify the rogue taxa, see Supplemental Material S4.

This study does not include the Pleistocene cricetine taxa, for which a revision beyond the scope of this paper is needed. The lack of the youngest cricetine fossil taxa from our analysis, makes the inferred position of the two included extant species uncertain. For this reason, they were removed from the resultant trees (Figs. 1, 2, S5.1–3) but are shown in Figs. S5.4–8 together with a discussion on the identified rogue taxa in Supplemental Material S5.

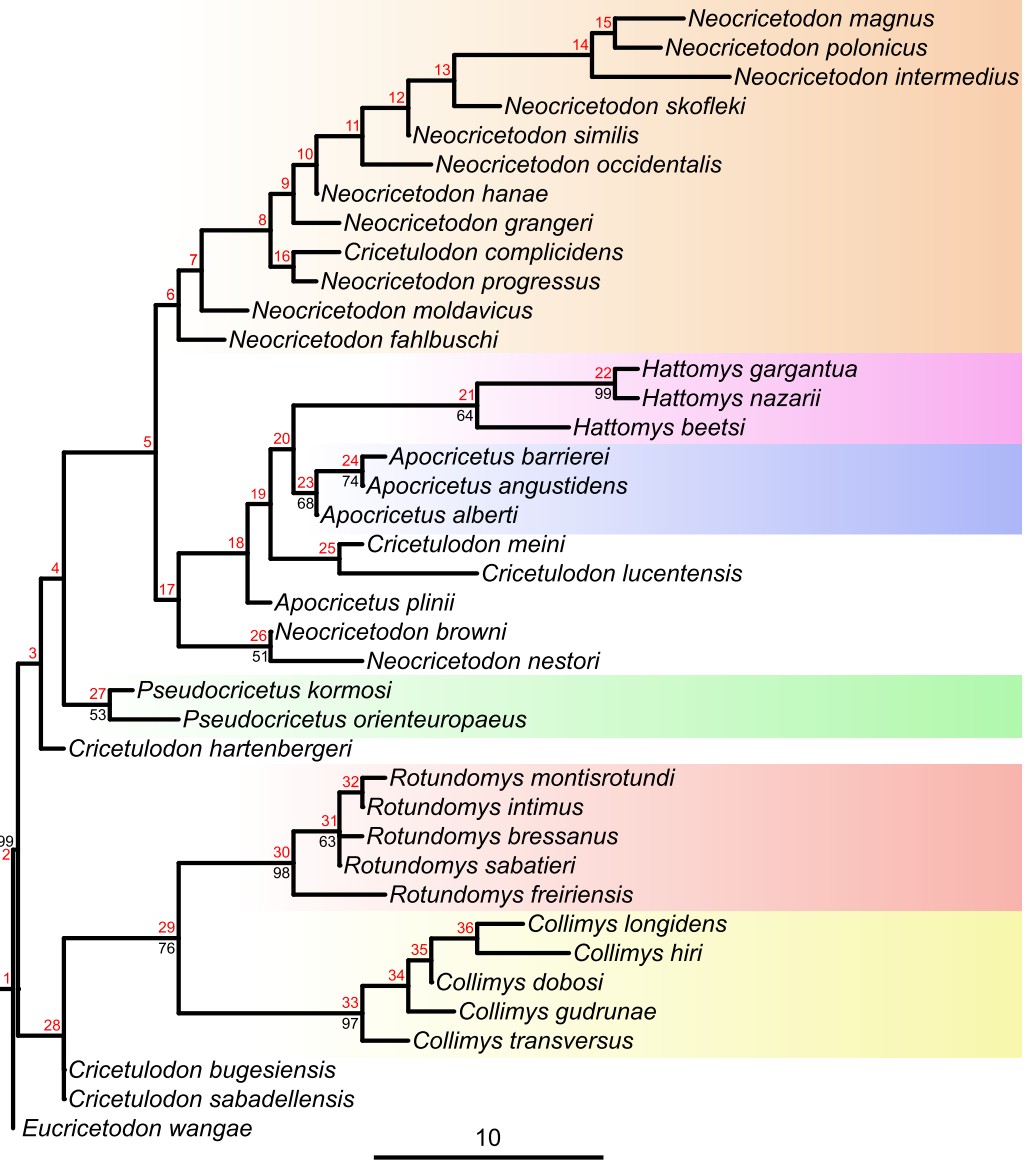

**Figure 2 Majority consensus tree (phylogram), calculated from the two most parsimonious trees of the implied weighting maximum parsimony analysis.** Bootstrap values over 50% are indicated at respective nodes in black, node numbers in red. The scale bar represents character state changes.

In total, the tree sets of all analyses were pruned by the same rogue taxa and the extant taxa, to obtain comparable trees. These trees, that are based on the pruned taxon set were then used for the subsequent analyses.

## Assessing stratigraphic congruence

To compare the fit of the topologies resulting from the different reconstruction methods, the R package strap v. 1.6 (*Bell & Lloyd, 2015*) was used to first time calibrate the non-clock trees, and then to calculate the following stratigraphic fit indices: (i) the relative completeness index (RCI) assesses the amount of gaps in the fossil record in relation to the

observed fossil ranges in the tree (*Benton & Storrs, 1994*); (ii) the gap excess ratio (GER) indicates the sum of ghost ranges in the tree scaled in relation to the possible minimum and maximum sum of ghost ranges in theoretical topologies (*Wills, 1999*); (iii) the modified Manhattan stratigraphic measure (MSM$^*$) indicates the sum of ghost ranges in the tree in relation to the sum of ghost ranges of the theoretical tree of best fit to the stratigraphy (*Siddall, 1998*; *Pol & Norell, 2001*). For all indices, significance tests were carried out, that resulted in very small *p*-values. Resulting indices of all analyses are listed in Supplemental Material S6.

## Character mapping

To ensure reliable results, two methods and two trees were used to map ancestral character states. Following the results of the stratigraphic congruence analyses, the implied weighting maximum parsimony tree was used to map characters in TNT under a parsimonious approach (see Supplemental Material S7) and the time-calibrated tree under the IGR clock model was used for stochastic character mapping with the R package phytools v. 2.1.1 (*Revell, 2024*). For the stochastic character mapping, three different models were fit to each character, with rates between states being either equal, symmetrical, or all different. The respective best fitting model was chosen using the Akaike information criterion and then used to map the character on the tree. Finally, for each character, the posterior probabilities of the different states were plotted on the nodes of the tree. For the reconstructed maps of characters, that are mentioned in the results, see Supplemental Material S8.

# RESULTS

Three taxa, *Apocricetus darderi*, *Neocricetodon ambarrensis* and *Pseudocricetus polgardiensis* were identified as rogue taxa, based on the clock trees and were pruned from the taxon set before calculating the consensus trees (MP-EW: 234 steps, CI: 0.453, RI: 0.757; MP-IW: 235 steps, CI: 0.451, RI: 0.755).

## Maximum parsimony analysis

Maximum parsimony analyses of morphological data sets including highly homoplastic characters can be improved by weighting characters according to their homoplasy (*Goloboff et al., 2008*). Consequently, implied weighting parsimony analyses can produce more resolved and accurate trees than standard equal weights (*Smith, 2019*). To assess which tree, from equal or implied weighting, fits better with the chronostratigraphy, stratigraphic congruence indices were calculated. While the topologies of both parsimony trees are quite similar (see Figs. 1 and S5.3), our results evidence a better stratigraphic congruence of the implied weighting tree than of the equal weighting tree (see Supplemental Material S5). Therefore, we discuss below the topology retrieved by applying implied weighting (Fig. 2).

   The topology of the consensus tree of the two most parsimonious trees shows a major basal split into two main clades. The first one (stemming from node 28) consists of *Collimys*, *Rotundomys*, *Cricetulodon bugesiensis* and *Cricetulodon sabadellensis* whereas the other one (stemming from node 3) includes the remaining cricetines. The most basal

taxon of this latter clade is *Cricetulodon hartenbergeri*, followed by *Pseudocricetus* (node 27), which is in turn sister to a large clade (node 5) that splits into two lineages. The first of them (stemming from node 6) includes *Cricetulodon complicidens* and most of the species belonging to the genus *Neocricetodon*. The second one (stemming from node 17) has as most basal taxa the sister species *N. browni* and *N. nestori*, as sister clade to the group stemming from node 18: *Apocricetus plinii*, followed by *Cricetulodon meini* plus *Cricetulodon lucentensis*, and the sister clades *Hattomys* and *Apocricetus sensu stricto* (*s.s.*).

## Tip-dated Bayesian analysis

The results of the stepping stone sampling strongly evidence a better fit for the 'fossiltip' sampling strategy under both clock models (2 $\log_e$(B10) > 10, see *Kass & Raftery, 1995*). Comparing Tk02 with IGR under 'fossiltip', the better fit model is the Tk02 (2 $\log_e(B_{10})$ = 4.6) but without strong evidence. However, the analysis under the Tk02 model showed some problems in reaching convergence (ESS below 200 for two parameters, 'convergence failed' according to Convenience). The IGR model showed a considerably better performance (lowest ESS > 3,300, 'convergence reached' according to Convenience). Moreover, the results of the strap analyses show a better stratigraphic congruence for the IGR model than for the TK02 model (see Supplemental Material S6). As both calibrated Bayesian inference analyses resulted practically in the same topology of the trees (see Figs. 2 and S5.2) with only slight differences in the posterior probabilities (generally a bit higher in Tk02) and in the relationships among the species belonging to *Neocricetodon* (particularly in that concerning the clades with low posterior probabilities), we describe below only the results obtained by applying the 'fossiltip' strategy under the IGR model (Fig. 3).

The topology of our tree evidences four major clades. The most basal one (stemming from node 35) consists of all species belonging to *Collimys*. Its sister clade (stemming from node 3) includes all remaining cricetines, with its most basal clade (stemming from node 31) including all *Rotundomys* species on the one side and the remaining two major clades (stemming from nodes 5 and 18) on the other. The first of the latter (stemming from node 5) consists of all species of *Neocricetodon*, as well as *Cricetulodon complicidens*. The second one (stemming from node 18) has *Cricetulodon hartenbergeri* as sister of two clades: a small one (node 30) including *Cricetulodon bugesiensis* and *Cricetulodon sabadellensis* and a larger one (stemming from node 19) constituted by a succession of clades with *Pseudocricetus* (node 29) at the base, followed by the sister taxa *Cricetulodon meini* and *Cricetulodon lucentensis* (node 28). One node up (node 22) inserts *A. plinii*, followed by the sister lineages *Apocricetus s.s.* and *Hattomys*. The topology of the tree supports the monophyly of *Collimys* (node 35), *Rotundomys* (node 31), *Neocricetodon* (node 5), *Pseudocricetus* (node 29), *Apocricetus s.s.* (node 26), and *Hattomys* (node 24). In contrast, the genus *Cricetulodon* is paraphyletic and only the type species *Cricetulodon sabadellensis* and *Cricetulodon bugesiensis* should be included in this genus.

**Divergence times.** The divergence times estimated using the two different clock models IGR and TK02, vary only slightly, with differences of mostly less than 500,000 years. In the same way, uncertainties on divergence times (measured by 95% highest probability

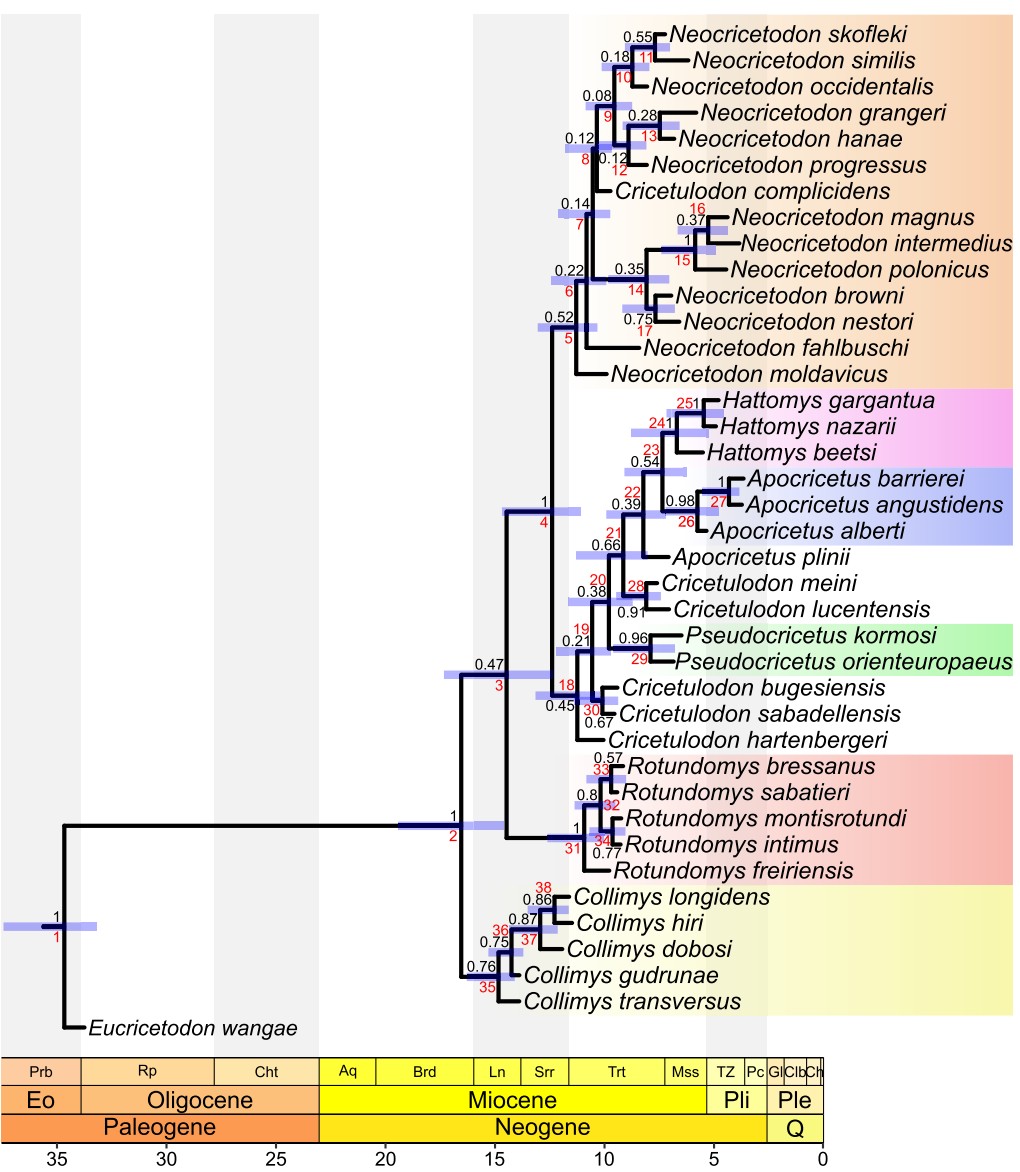

**Figure 3 MCC tree of the time-calibrated relaxed-clock IGR Bayesian inference analysis.** Posterior probabilities of clades are indicated at respective nodes in black, node numbers in red, node bars indicate the 95% highest posterior density for divergence times. The scale axis is in Ma, the chronostratigraphic chart follows *Cohen, Harper & Gibbard (2022)*.

densities (HPD) ranges) are quite similar independently of the clock model applied. In the following, the results of the analysis under the IGR model are reported. They reveal a late Early Miocene age for the first split within the ingroup (16.54 Ma, 95% HPD: 14.53–19.12 Ma). *Collimys* is recovered as the oldest genus, diverging during the Middle Miocene (14.81 Ma, 95% HPD: 14.07–15.85 Ma). The remaining five monophyletic genera diverged later, during the Late Miocene: *Neocricetodon* (11.28 Ma, 95% HPD: 10.26–12.64 Ma), *Rotundomys* (10.91 Ma, 95% HPD: 10.01–12.23 Ma), *Pseudocricetus* (7.9 Ma, 95% HPD:

6.69–9.29 Ma), *Hattomys* (6.7 Ma, 95% HPD: 5.17–8.47 Ma), and *Apocricetus s.s.* (5.7 Ma, 95% HPD: 4.59–6.81 Ma).

### Ancestral character state reconstructions

The slight differences in the topology of both trees (particularly regarding the phylogenetic position of *Rotundomys* and *Collimys*) result in different reconstructed synapomorphies for some clades. However, concerning the genera, most of the synapomorphies found when analysing the results of the stochastic character mapping are also retrieved by the parsimonious mapping of synapomorphies (see Supplemental Materials S7 and S8). Only a few synapomorphies are not found with the latter approach, due to the coding of polymorphic states as 'ambiguous' in the mapping process in TNT. Therefore, below we list mainly the synapomorphies obtained by applying stochastic character mapping.

All species belonging to *Collimys* (stemming from node 35) share the exclusive synapomorphy of having an ectomesolophid on the m1 (68: 0→1). Additional ambiguous synapomorphies are: *e.g.*, a labial spur of the anterolophule reaching the labial border of the M1 (103: 0→2), a long mesoloph on the M3 (49: 2→0) and the presence of a labial spur of the anterolophulid on the m1 (64: 0→1).

The clade *Rotundomys* (stemming from node 31) is defined by the following exclusive synapomorphies: a very weak to absent mesoloph on M1 and M2 (20: 1→2, 37: 1→2), a posteroloph that is merged with the posterior metalophule on the M1 (26: 1→3) and the presence of a short or hanging labial anterolophid on the m3 (88: 0→1), as well as many non-exclusive synapomorphies.

Representatives of the clade stemming from node 5 including *Neocricetodon* and *Cricetulodon complicidens* share non-exclusive synapomorphies such as the presence of the anterior metalophule on the M2 (38: 2→1/0). Additional non-exclusive synapomorphies, only shared with *Collimys*, are the presence of a long mesolophid on the m1 (66: 0/1→2) and having a medium to long labial spur of the anterolophule on the M1 that reaches the molar border in most of the taxa belonging to *Neocricetodon* (103: 0→1/2).

Representatives of *Pseudocricetus* (stemming from node 29) are characterised by having a short but distinct posteroloph on the M3 (56: 0→1) and the tendency to form a very small mesolophid on the m2 (73: 2→1).

*Apocricetus s.s.* (stemming from node 26) is characterised by the two non-exclusive synapomorphies of having a multi-lobed, crestiform anteroconid (57: 2→4) with a poorly developed labial anterolophid (60: 0→1) on the m1.

Representatives of *Hattomys* (stemming from node 24) are clearly distinct from those belonging to its sister group *Apocricetus s.s.* They are characterised by sharing the two exclusive synapomorphies of having large M1, longer than 3.2 mm (1: 1→3) and an hypolophulid connected to a medium sized mesolophid on the m1 (109: 2→5).

## DISCUSSION

### Maximum parsimony *vs.* Bayesian trees

All maximum Parsimony and Bayesian tip-dated and undated trees (Figs. 1, 2, S5.1–3) support the monophyly of *Collimys*, *Rotundomys*, *Pseudocricetus Apocricetus s.s.* and

*Hattomys*. *Neocricetodon* includes *Cricetulodon complicidens* in all trees. It is monophyletic in the Bayesian trees but not in the parsimonious ones, as two species branch outside of the clade (*N. browni* and *N. nestori* in the implied weighting analysis and *N. moldavicus* and *N. fahlbuschi* in the equal weighting one).

*Cricetulodon* splits up in the same groups in both, the IGR Bayesian clock analysis and the implied weighting parsimony analysis (Figs. 1 and 2). However, some differences are observed concerning the phylogenetic position of the clades. In this way, the topology of the parsimony tree shows *Cricetulodon bugesiensis* and *Cricetulodon sabadellensis* as the most basal taxa of the clade including *Collimys* and *Rotundomys* (stemming from node 28, Fig. 2), whereas in the Bayesian topology this lineage belongs to the clade that comprises the remaining species of *Cricetulodon*, *Pseudocricetus*, *Apocricetus* and *Hattomys* (stemming from node 18, Fig. 3). Moreover, *A. plinii* is basal to *Cricetulodon meini* and *Cricetulodon lucentensis* in the parsimony tree, but sister to *Apocricetus s.s.* and *Hattomys* in the Bayesian one (inserting at node 18, Fig. 2 *vs.* node 22, Fig. 3).

The most striking difference when comparing the Bayesian clock tree to the parsimony one concerns the relationship between *Collimys* and *Rotundomys*, for which the parsimony analysis found a sister relationship (see node 29, Fig. 2, BS = 76) whereas in the Bayesian tree they insert sequentially (at node 2 and 3, Fig. 3). Therefore, depending on the topology, contrasting ancestral character state reconstructions are found. The characters that are different in *Collimys* and *Rotundomys* compared to the remaining cricetines are the following: M1 anterocone not clearly divided, but more crestiform *vs.* divided in two (5: 6 *vs.* 2); M1 lingual anteroloph clearly present *vs.* weak or absent (11: 0 *vs.* 1); upper molars protolophule posterior *vs.* double (7: 0 *vs.* 1, 34: 2 *vs.* 1, 47: 3 *vs.* 0); m3 lingual anterolophid well-developed *vs.* weak or absent (76: 1 *vs.* 0, reversed in *Apocricetus s.s.* and *Hattomys* (76: 0 *vs.* 1)). According to the stochastic character mapping, all these characters serve as synapomorphies for the clade *Neocricetodon* + *Cricetulodon* + *Pseudocricetus* + *Apocricetus* + *Hattomys* (stemming from node 4, Fig. 3). In the parsimony analysis, on the other side, the results are reversed. Only having a double protolophule on the M1 is reconstructed as a synapomorphy for this clade (stemming from node 3, Fig. 2), while the other above-mentioned characters serve here as synapomorphies for the clade *Collimys* + *Rotundomys* (node 29, Fig. 2).

The genera *Rotundomys* and *Collimys* were informally grouped by Kälin (1999) on the basis of emerging hypsodonty, whereas Heissig (1995) excluded *Collimys* as potential ancestor of *Rotundomys*, stating that the tendency of acquiring hypsodonty evolved independently.

In the parsimony analysis, *Cricetulodon sabadellensis* and *Cricetulodon bugesiensis* are additionally positioned as sister taxa to *Collimys* and *Rotundomys*. Considering the much younger age of the *Cricetulodon* taxa compared to *Collimys*, the arrangement in the Bayesian tree seems more plausible.

### Collimys

According to our results, *Collimys* forms a monophyletic group, from which *Collimys transversus* and *Collimys gudrunae* are splitting first. Prieto & Rummel (2009a, 2009b)

proposed three different lineages within this genus. An early lineage *Collimys transversus–Collimys gudrunae*, a second temporally intermediate lineage involving *Collimys* sp. 1, 2 (from Petersbuch 10, 18, 6 and 48) and a later lineage *Collimys hiri–Collimys longidens–Collimys dobosi*. Our results, nevertheless, do not support the lineages proposed by *Prieto & Rummel (2009a, 2009b)*. Moreover, according to the topology of our tree, the clade *Collimys dobosi* plus more derived taxa (stemming from node 37, Fig. 3) differs significantly from the *Collimys hiri–Collimys longidens–Collimys dobosi* lineage proposed by *Prieto & Rummel (2009a, 2009b)*. The latter authors proposed the lineage mainly on the basis of an increase in size, in hypsodonty and in mesoloph length on the M1 and M2, together with other minor morphological variations but the validity of the lineages has been doubted later, due to regional variation of the taxa (*Prieto et al., 2014*; *Hír et al., 2017*). Our results show that the most basal position of *Collimys transversus* and *Collimys gudrunae* is supported by the smaller size of *Collimys transversus* and the presence of a slightly more developed lingual anteroloph on the M3, in both reconstructed phylogenies. *Collimys dobosi* is the next splitting species, sister to *Collimys hiri* and *Collimys longidens*, due to a more square-shaped M2, which is elongated in the latter taxa, and the presence of an anterior metalophule in the M3. The absolute size differences between the species are too small to result in different states in the phylogenetic matrix. The length of the mesoloph on the M1 is variable in *Collimys dobosi, Collimys hiri* and *Collimys longidens* but it does not reach the border of the teeth in the majority of the specimens, whereas it usually does on the M2 of all three taxa (*Kälin & Engesser, 2001*; *Hír, 2005*; *Prieto & Rummel, 2009b*). Therefore, the reconstructions resulted in a more basal *Collimys dobosi*, in both dated and undated analyses, due to above mentioned reasons. This basal position of *Collimys dobosi* compared to *Collimys hiri* and *Collimys longidens* is congruent to the slightly older age attributed to the former species. It has been recorded from Felsőtárkány, Hungary (~12.2–11.6 Ma), while *Collimys longidens* and *Collimys hiri* have been found in Nebelbergweg, Switzerland and Hammerschmiede, Germany, respectively (~11.9–11.3 Ma) (*Hír et al., 2016, 2017*; *Prieto & Rummel, 2016*).

### *Rotundomys*

*Rotundomys freiriensis* as the basal-most taxon within *Rotundomys* is the best supported split in the reconstructed phylogenies. It is based on the absence of the mesoloph (or anterior metalophule) on the M3, which is present in the remaining species of the genus and in having the lingual anterolophid on m1 and m2 better developed than in more derived taxa. This arrangement follows the proposal of several previous studies (*Antunes & Mein, 1979*; *Freudenthal, Mein & Martín Suárez, 1998*; *López-Antoñanzas, Peláez-Campomanes & Álvarez-Sierra, 2014*). *Rotundomys montisrotundi* and *R. intimus* form a clade characterised by a poorly developed lingual anteroloph on the M2. *López-Antoñanzas, Peláez-Campomanes & Álvarez-Sierra (2014)* addressed the similarity between these two species and with *R. sabatieri*, and separated them on the basis of size, as well as some minor morphological differences and different proportions of morphotypes. The results of their phylogenetic analysis showed *R. montisrotundi* and *R. bressanus* as sister species, based on the absence of the lingual anteroloph on the M2 in most specimens.

However, their phylogenetic analysis only included the genera *Rotundomys* and *Cricetulodon* and therefore these results should be taken with caution before drawing general conclusions. *Mein (1975)* proposed *R. bressanus* to be derived from *R. montisrotundi*, mainly based on size differences. Intraspecific variation of *R. montisrotundi* complicates, however, confirming or refuting this hypothesis (*Freudenthal, Mein & Martín Suárez, 1998*).

In general, discrimination between species often relies on differences in size, that can, however, overlap in their ranges (see *R. sabatieri vs. R. bressanus* in *Aguilar, Michaux & Lazzari, 2007*), resulting in short branch lengths in the maximum parsimony phylogram and in collapsed clades, due to zero-size branch lengths (see *R. sabatieri*). Consequently, the only reliable relationships within the genus seem to be the basal position of *R. freiriensis* and the closely related *R. montisrotundi* and *R. intimus*.

### Neocricetodon

The name *Neocricetodon Schaub, 1934* was validated by *Daxner-Höck et al. (1996)* and followed by a majority of authors afterwards.

Our results show three synapomorphies for *Neocricetodon*: (i) the presence of a long mesolophid on the m1, (ii) the presence of a labial spur of the anterolophule on the M1 and (iii) the presence of an anterior metalophule on the M2. These results agree with those of *Freudenthal, Mein & Martín Suárez (1998)*, regarding the synapomorphies (i) and (ii). Additionally, they also mentioned that the species belonging to *Neocricetodon* show a labial anterolophulid on the m1 and elongated mesolophs on the upper molars. Our results are also in line with those of *Sinitsa & Delinschi (2016)*, agreeing on the synapomorphy (iii) the presence of the anterior metalophule on the M2. The latter authors additionally proposed as synapomorphies of this clade an expanded anterocone and the presence of a labial anterolophule on the M1, a labial anterolophulid on the m1, and a four rooted M2. They have, however, only included *Cricetulodon sabadellensis*, '*Kowalskia* cf. *schaubi*' (*Kretzoi, 1951*) and *Democricetodon* as outgroup taxa but no other cricetines, therefore these proposed synapomorphies must be treated carefully. A four rooted M2, for example, is reconstructed as plesiomorphic for *Neocricetodon* by the stochastic character mapping.

Interestingly, the first two synapomorphies we have proposed for *Neocricetodon*, (i) the presence of a mesolophid on the m1 (reaching the molar border in most of the taxa) and (ii) the presence of a labial spur of the anterolophule on the M1 (reaching the border of the molar in most taxa) are not considered as synapomorphies by *Sinitsa & Delinschi (2016)*. In fact, these authors, coded *N. occidentalis*, *N. progressus* and *N. moldavicus* as lacking or having a short mesolophid on the m1 (*Sinitsa & Delinschi, 2016*; Table 2). However, previous studies have described *N. occidentalis* and *N. progressus* as having usually long mesolophids (*de Bruijn et al., 1975*; *Freudenthal, Lacomba & Martín Suárez, 1991*; *Topachevsky & Skorik, 1992*; *Freudenthal, Mein & Martín Suárez, 1998*; *Sinitsa, 2012*) and *N. moldavicus* as having short to medium mesolophids. Moreover, the holotype of this latter species shows a clearly well-developed mesolophid (*Lungu, 1981*; *Sinitsa & Delinschi, 2016*). Regarding (ii) the spur of the anterolophule on the M1, *Sinitsa & Delinschi (2016)* coded having short or absent spurs as one single state of character. This could be the reason

why the presence of this structure has not been expressed in their matrix of characters in several taxa (*e.g.*, *N. nestori*, see *Engesser, 1989*), impeding its identification as a possible synapomorphy.

Both the above-mentioned structures, the mesolophid on the m1 and the labial spur of the anterolophule on the M1, are relatively poorly developed in *N. moldavicus*, justifying its basal position within the clade. Only the third synapomorphy, (iii) the presence of the anterior metalophule on the M2 (coded here as metalophule either anterior or double), is clearly observable in *N. moldavicus*. However, as noticed by *Freudenthal, Mein & Martín Suárez (1998)*, who did not include this character as a diagnostic trait of the genus, it is quite variable in several taxa (*e.g.*, *N. nestori*, *N. progressus*, *N. hanae*, *N. browni*), although there is a strong tendency towards its presence. *Sinitsa & Delinschi (2016)* termed the character as 'phylogenetically irrelevant', due to homoplasy and reversals in some clades. They specifically mentioned the loss of this structure in *N. grangeri* but the single M2 from the original material is too heavily damaged to make any statement concerning the metalophule (*Daxner-Höck et al., 1996*). Yet, additional found material of this species, that includes several complete M2s, evidences that most of them have an anterior metalophule (*Wu & Flynn, 2017*).

Several phylogenetic hypotheses within *Neocricetodon* have been proposed (*Wu, 1991*; *Daxner-Höck, 1992*; *Freudenthal, Mein & Martín Suárez, 1998*; *Qiu & Li, 2016*; *Sinitsa & Delinschi, 2016*). However, the evolutionary history of this taxon, that includes numerous species with wide geographical and temporal distribution, is complex to untangle. This complexity is also reflected in our results, which put in evidence low posterior probabilities for most of the clades within *Neocricetodon* and some differences in the topologies of the trees, which mostly prevents reliable statements about the proposed lineages. However, the clade combining *N. magnus*, *N. intermedius* and *N. polonicus* (stemming from node 14, Fig. 2, or node 15, Fig. 3), which were dominantly distributed in Eastern Europe, in Hungary, Poland, Slovakia and Ukraine, (*Fahlbusch, 1969*; *Jánossy & Kordos, 1977*; *Pevzner et al., 1996*) is consistent in all our analyses and very well supported in the IGR Bayesian tree (PP = 1). This result is in disagreement with the hypothesis of *Wu (1991)*, who separated the three species into three different lineages mainly on the basis of their molar size. Conversely, it corroborates the close relationship between *N. polonicus* and *N. intermedius*, as recovered by *Sinitsa & Delinschi (2016)*, who did not include *N. magnus* in their analysis.

According to the topology of our tree, *Neocricetodon* also includes *Cricetulodon complicidens*. Difficulties regarding the genus assignment of this species were already mentioned in several papers (*Freudenthal, Mein & Martín Suárez, 1998*; *Kälin, 1999*). *Topachevsky & Skorik (1992)*, who coined this species, described some similarities with *Neocricetodon*, such as the presence of a long mesolophid on the m1, a long labial spur of the anterolophule on the M1, and an anterior metalophule on the M2. These are, in fact, the three above mentioned synapomorphies that define this group. Consequently, the reallocation of *Cricetulodon complicidens* into the genus *Neocricetodon* seems to be justified.

This reallocation could seem to cause confusion considering another species coined by *Topachevsky & Skorik (1992)* as 'Kowalskia complicidens' due to the fact that *Kowalskia* is considered a junior synonym of *Neocricetodon* by several authors (*Freudenthal, Mein & Martín Suárez, 1998*; *Sinitsa & Delinschi, 2016*). However, 'Kowalskia' complicidens is thought not to belong to *Neocricetodon* (or *Kowalskia*) but rather to *Sinocricetus Schaub, 1930* (*Daxner-Höck et al., 1996*; *Qiu & Li, 2016*; *Sinitsa & Delinschi, 2016*).

This leads to the question about the validity of the genus *Kowalskia* or its synonymy with *Neocricetodon*. The scarce material of the type species *N. grangeri* did not help to clarify this issue and therefore some authors keep the genera separated (*Daxner-Höck et al., 1996*), whereas others consider *Kowalskia* as junior synonym of *Neocricetodon*, until the discovery of additional material of this species would allow to either detect clear similarities or differences (*Freudenthal, Mein & Martín Suárez, 1998*). *Sinitsa & Delinschi (2016)* reconstructed the phylogeny of the group. Their work retrieved *K. polonica*, the type species of *Kowalskia*, branching in the same clade as *N. grangeri* and other species of *Neocricetodon*. However, due to the limited material of *N. grangeri*, they could not code any of the characters related to the M1 of this species, which make up nearly half of the total characters of their matrix. Soon afterwards, *Wu & Flynn (2017)* published additional material of *N. grangeri*, which helped them to conclude that the synonymy of *Kowalskia* with *Neocricetodon* is strongly supported.

The topologies of the parsimony, undated and tip dating TK02 clock trees (Figs. 1, S5.1, 2) agree with *Sinitsa & Delinschi (2016)* in the phylogenetic position of the type species of *Kowalskia* and *Neocricetodon*, as nesting within a clade that includes the remaining species of *Neocricetodon* (stemming from node 6, Fig. 2). All these trees evidence a derived position of *K. polonica*, which shares the three synapomorphies mentioned above that characterise the genus *Neocricetodon*. Only the Bayesian IGR clock tree (Fig. 3) shows two main clades inside the clade *Neocricetodon*, one of which (stemming from node 8, Fig. 3) includes the type species *N. grangeri* and the other one (stemming from node 14, Fig. 3) includes *K. polonica*. These two clades could be interpreted as two separated genera, with all species in the same clade as *K. polonica* reallocated to *Kowalskia*. However, taking into account the very low posterior probabilities of these clades (0.12 and 0.35), the absence of clear synapomorphies for both of the clades, and that these clades are not recovered in any of the remaining analyses presented here, we consider the synonymy of *Kowalskia* with *Neocricetodon* to be justified.

### *Cricetulodon*

The difficulty of defining the genus *Cricetulodon* is exemplified by previous proposals of synonymising it with *Rotundomys* or with *Neocricetodon* (*Freudenthal, 1967*, *1985*). *Freudenthal, Mein & Martín Suárez (1998)* eventually separated *Cricetulodon* from *Neocricetodon* mainly based on the presence of a mostly lingual anterolophulid on the m1 of the former taxon. Due to the high variability observed on the anterior part of the m1 of these two taxa, the determination of a dominantly lingual or labial anterolophulid can be problematic, particularly, when the anterolophid is double or more centrally positioned (*Engesser, 1989*; *Wu, 1991*; *Daxner-Höck & Höck, 2015*). The problems of relying on this

variable character to allocate a species into a genus are exemplified by the above-mentioned *Cricetulodon complicidens*.

The topology of our trees (Figs. 1 and 2) does not support the monophyly of *Cricetulodon*, which is in agreement with previous phylogenetic studies (*López-Antoñanzas, Peláez-Campomanes & Álvarez-Sierra, 2014*). However, the work of these authors was focused on a new species of *Rotundomys* and only the species belonging to *Rotundomys* and *Cricetulodon* were analysed. The three clades they recovered, *Cricetulodon hartenbergeri* plus *Cricetulodon sabadellensis*, *Cricetulodon bugesiensis* plus *Cricetulodon lucentensis*, and *Cricetulodon meini*, all basal to *Rotundomys*, are not found in our trees. Our results show *Cricetulodon meini* and *Cricetulodon lucentensis* as sister species, which is consistent with the hypothesis of a potential ancestor-descendant relationship between these two taxa suggested by *Freudenthal, Mein & Martín Suárez (1998)*. After *Freudenthal (1967)*, the position of *Cricetulodon hartenbergeri* and *Cricetulodon sabadellensis* as potential ancestors of *Rotundomys*, was adopted and discussed by several authors (*Fejfar, 1970*; *Daxner-Höck, 1972*; *Kälin, 1999*; *Fejfar et al., 2011*; *López-Antoñanzas, Peláez-Campomanes & Álvarez-Sierra, 2014*). While *Cricetulodon sabadellensis* and *Cricetulodon bugesiensis* are recovered as possible ancestors of *Rotundomys* in the undated analyses (see node 28, Fig. 2), they are quite distant in the clock trees, which is likely resulting from their similar or even younger age compared to *R. freiriensis*.

Be that as it may, the clade consisting of *Cricetulodon lucentensis* and *Cricetulodon meini* is not closely related to the type species of the genus *Cricetulodon sabadellensis* in any of our trees. Consequently, these species should be excluded from the genus and transferred into a new one.

### *Pseudocricetus*

This genus was partly defined on the basis of some characters of the mandible, the skull and the incisors by *Topachevsky & Skorik (1992)* and *Sinitsa (2010)*. Some dental morphological characters, such as the presence of reduced mesolophs and mesolophids, anterior protolophules or the split of the anterocone that were proposed to define *Pseudocricetus*, are in fact also present in several other genera (*Daxner-Höck et al., 1996*; *Freudenthal, Mein & Martín Suárez, 1998*). According to our results, *Pseudocricetus* is monophyletic and, in the Bayesian tree, sister clade to the lineage of *Cricetulodon lucentensis*, *Cricetulodon meini*, *Apocricetus* and *Hattomys* (stemming from node 21, Fig. 3). This agrees with previous hypotheses, according to which there were morphological similarities between *Pseudocricetus* and *Apocricetus* (*Kälin, 1999*), or that considered *Pseudocricetus* as a possible ancestor of *Hattomys* (*Freudenthal & Martín Suárez, 2010*).

### *Apocricetus*

*Freudenthal, Mein & Martín Suárez (1998)* proposed the phyletic lineage, *Apocricetus plinii*–*A. alberti*–*A. barrierei*–*A. angustidens*. According to these authors, the changes

along this lineage, *e.g.*, the development of the anterior protolophules or the presence of an anterior ridge in the M1, are gradual and refer to size as well as to morphological features (see also *Ruiz-Sánchez et al., 2014*; *Mansino et al., 2014*). The topologies of our trees mostly agree with the thoughts of *Freudenthal, Mein & Martín Suárez (1998)* except for *A. plinii*, which, despite being basal to *Apocricetus s.s.*, does not belong to this clade. Instead, the results of the Bayesian analysis show *A. plinii* (inserting at node 22, Fig. 3) as sister species to the sister clades *Apocricetus s.s.* and *Hattomys*. The topology of the maximum parsimony tree shows *A. plinii* (inserting at node 18, Fig. 2) as basal to the clade (stemming from node 19, Fig. 2) consisting in *Cricetulodon meini* plus *Cricetulodon lucentensis* and the sister clades *Apocricetus s.s.* and *Hattomys*.

*Apocricetus plinii* and *A. alberti* show less derived features such as a better developed anterior protolophule on the M1 when comparing with *A. angustidens* and *A. barrierei*. However, *A. plinii* differs from *A. alberti* by its less derived morphology of the anteroconid on the m1, which is not crest-like but split into two anteroconids. In addition, the labial spur of the anterolophule on the M1 of *A. plinii* is usually free and not connected to the labial anterocone as is the case of *A. alberti* (*Freudenthal, Mein & Martín Suárez, 1998*). Therefore the 'anterior atoll' that is formed between the two anterocones in all species belonging to *Apocricetus s.s.* and *Hattomys*, is often absent in *A. plinii*, which could explain its phylogenetical position in the tree.

### *Hattomys*

Regarding the characters that have been used to define *Hattomys*, special attention was paid to the mesoloph(id) and the so-called 'preloph(id)' (*Freudenthal, 1985*; *Savorelli, 2013*). Due to the direction and position of the structure that connects the ectoloph and the entoconid, it is difficult to know whether it represents the mesolophid, the anterior hypolophulid or a combination of both (*Freudenthal, 1985*; *Savorelli, 2013*). This structure is here interpreted as a mesolophid of medium length (or long in the case of the m3), which is fused with the anterior hypolophulid to some extent (well visible in *Savorelli, 2013*, Fig. 5.6). The so-called 'prelophid' is here interpreted as a lingual spur of the anterolophulid that is connected to the anterior metalophulid and to the posterior spur of the lingual anteroconid. In the upper molars, the 'preloph' is, accordingly, the labial spur of the anteroloph, which is either connected to a longitudinal running anterior protolophule and the posterior spur of the labial anterocone, or runs freely, as frequently seen in *H. beetsi*. This spur can also continue towards the labial border after its connection to the labial anterocone (see *e.g.*, *Freudenthal, 1985*, plate 3.1). On all upper molars, the anterior metalophule seems to be lacking and there is only a long mesoloph, that usually reaches the labial border of the tooth. It can sometimes connect to the metacone.

The presence of a 'preloph(id)' is, independently of its interpretation, not a synapomorphy for *Hattomys* considering its frequent presence in *Apocricetus s.s.* (*Ruiz-Sánchez et al., 2014*; *Mansino et al., 2014*). Instead, two non-exclusive synapomorphies, the presence of mesolophids of medium length on the m1 and the m2, and a single exclusive synapomorphy, the connection of the hypolophulid to the mesolophid on the m1, are

identified by the stochastic character mapping. Moreover, synapomorphies proposed in previous studies (*Freudenthal, 1985*; *Savorelli, 2013*), such as the presence of a long mesoloph on the M2 and the M3 that reaches the labial border of the tooth and the well-developed 'flanges' on the cusps were only identified in *H. nazarii* and *H. gargantua*, here. These characters turned out to be plesiomorphic in *H. beetsi*, which could explain the basal-most position of this taxon inside the clade.

*Hattomys* is found in Gargano peninsula, Italy, as part of a clearly insular fauna. There is uncertainty regarding the timing and modes of colonisation of the island (*Mazza & Rustioni, 2008*; *van den Hoek Ostende, Meijer & van der Geer, 2009*; *Freudenthal & Martín Suárez, 2010*; *Freudenthal, van den Hoek Ostende & Martín-Suárez, 2013*; *Savorelli & Masini, 2016*). Due to the uncertainty of the age of the fauna, a relatively large interval of time was chosen for these three species, regarding the tip-dating. This could explain the relatively large 95% HPD range of the estimated divergence time of this clade, when compared with other taxa such as *Apocricetus s.s.*

Possible ancestors of *Hattomys* were assumed to be found in *Neocricetodon*, *Pseudocricetus* or *Apocricetus* (*Freudenthal & Martín Suárez, 2010*). *Freudenthal (1985)* especially emphasized the similarity between *Hattomys* and *A. alberti*, which is congruent with the close relationship between *Hattomys* and *Apocricetus s.s.*, that we retrieved in this study. According to the topology of our Bayesian tree, the common ancestor of *Hattomys* and *Apocricetus s.s.* could be a species close to *A. plinii*. The timing of the split between these two genera (7.41 Ma, 95% HPD: 6.23–8.8 Ma), and to *A. plinii* (8.22 Ma, 95% HPD: 7.1–9.56 Ma) could have an impact in the estimations of the age of the Gargano fauna. *Freudenthal, van den Hoek Ostende & Martín-Suárez (2013)* assume a single colonisation event at around 8.8–7.5 Ma, which fits quite well with the here reconstructed divergence estimations.

## CONCLUSION

This study is the first to analyse the origin and early diversification of cricetine rodents based on a morphological only dataset of late Miocene and Pliocene fossils applying Bayesian and parsimony methods. Our results unravel the relationships within and between several of its genera, providing answers to their systematic uncertainties. This work evidences that the genera *Collimys*, *Rotundomys*, *Pseudocricetus*, *Apocricetus s.s.* and *Hattomys* are monophyletic whereas *Cricetulodon* is paraphyletic. The species *Apocricetus plinii* does probably not belong to *Apocricetus*, being basal to the sister clades *Apocricetus s.s.* and *Hattomys*. *Pseudocricetus* is closer to the *Apocricetus-Hattomys* clade than to *Neocricetodon*. Finally, *Kowalskia* is confirmed as a junior synonym of *Neocricetodon* with 'Cricetulodon' complicidens being most likely a member of this genus. The new insights into the relationships between these extinct genera, help to gain a better understanding of the evolutionary history of the Cricetidae. Based on the expanded morphological matrix, additional extinct and also extant members of the group can be rapidly added to the phylogeny in future studies. Hence, this work provides the first basis for the still relatively poorly understood origin of today's hamsters.

# ACKNOWLEDGEMENTS

We are grateful to Anne-Lise Charruault (ISEM, Univ Montpellier), Emmanuel Robert (lgl tpe UCB Lyon 1), Gertrud Rößner (SNSB-BSPG, Munich) and Imelda Hausmann (SNSB-BSPG, Munich) for access to the specimens of the University of Montpellier, the FSL and the SNSB-BSPG, as well as to Jonathan Mitchell (Coe College, Cedar Rapids) for helping with the R code to reconstruct the ancestral character states.

### Funding

This research was funded by the research project ANR22-CE02-0022 (MESRI/ANR) and the project PID2023-151089NB-I00 funded by MCIU/AEI/10.13039/501100011033/ FEDER, UE. The funders had no role in study design, data collection and analysis, decision to publish, or preparation of the manuscript.

### Grant Disclosures

The following grant information was disclosed by the authors:
ANR22-CE02-0022 (MESRI/ANR).
PID2023-151089NB-I00 and MCIU/AEI/10.13039/501100011033/FEDER, UE.

### Competing Interests

Raquel López-Antoñanzas is an Academic Editor for PeerJ.

### Author Contributions

- Moritz Dirnberger conceived and designed the experiments, performed the experiments, analyzed the data, prepared figures and/or tables, authored or reviewed drafts of the article, and approved the final draft.
- Pablo Peláez-Campomanes conceived and designed the experiments, analyzed the data, authored or reviewed drafts of the article, and approved the final draft.
- Raquel López-Antoñanzas conceived and designed the experiments, analyzed the data, authored or reviewed drafts of the article, and approved the final draft.

### Data Availability

The morphological matrix on which the phylogenetic reconstructions are based, the input and output files of the stepping stone sampling and the phylogenetic reconstructions in MrBayes and TNT, the R code used for the identification of rogue taxa, and the assessment of the stratigraphic fit of the tree topologies and the stochastic character mapping are available in the Supplemental File.

### Supplemental Information

Supplemental information for this article can be found online at http://dx.doi.org/10.7717/ peerj.18440#supplemental-information.

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
