# Peer review of "Phylogenetic relationships of Neogene hamsters (Mammalia, Rodentia, Cricetinae) revealed under Bayesian inference and maximum parsimony"

_PeerJ, doi:10.7717/peerj.18440_

## Round 0.1 · original submission · Minor Revisions

Please, address the reviewers' comments and suggestions.

·

Basic reporting

The manuscript is written clearly and concisely in correct English. The references are appropriate and relevant, supporting the study effectively. The structure adheres to the format required by PeerJ, with appropriate content in each section. The introduction provides a comprehensive context for the study. The figures presented are relevant and well-labelled, offering clear visual support to the text.

Experimental design

This research presents an original investigation with a well-defined question. The methods are meticulously described and well-defined, providing a solid foundation for reproducibility. The study utilises traditional maximum parsimony and Bayesian inference approaches, ensuring robustness in the analysis. The inclusion of a relaxed-clock Bayesian inference analysis to estimate divergence times is a notable strength.

Validity of the findings

The findings are valid and contribute significantly to the understanding of phylogenetic relationships among Neogene hamsters. The study successfully confirms the monophyly of most genera and identifies synapomorphies based on ancestral state reconstructions. The treatment of rogue taxa enhances the informative value of the phylogenetic trees. The identification of Cricetulodon as paraphyletic, with Cricetulodon complicidens classified under Neocricetodon, is a significant outcome that aligns well with the study's goals.

Additional comments

The manuscript titled "Phylogenetic relationships of Neogene hamsters (Mammalia, Rodentia, Cricetinae) revealed under Bayesian inference and maximum parsimony" presents noteworthy findings that substantially contribute to the understanding of Cricetinae evolution. The study's results pave the way for new discussions and set a strong foundation for future research. However, I include below some comments that should be considered by the authors as well as some suggestions on the manuscript pdf included as an attached file. The suggested revisions aim to enhance clarity, depth, and robustness, ensuring that the study's conclusions are well-supported and informative:

Intraspecific Variability:
Discussion on Intraspecific Variability: The manuscript briefly mentions handling intraspecific variability by considering only the condition found at the type locality (line 113). However, it is essential to explore this aspect further in the results and conclusions. Intraspecific variability can significantly impact phylogenetic analyses, especially when dealing with morphological characters. Previous studies have highlighted the importance of accounting for such variability. For instance, Zanesco et al. (2019) and Hayden et al. (2023) emphasised that dental characters exhibit significant variability even within a species. This variability can influence the topology of phylogenetic trees if multiple specimens per species are included. Therefore, a detailed evaluation of how intraspecific variability might affect the study's findings is crucial (Brocklehurst N, Benevento GL. 2020).
Impact on Dental Characters: Dental characters, being crucial for phylogenetic studies in rodents, exhibit significant developmental variability. Hayden et al. (2023) explored this aspect in mouse molars, revealing substantial morphological differences even within genetically uniform populations. Such variability underscores the complexity of dental character evolution and the necessity to consider these factors in phylogenetic studies. Addressing these points will add depth to the manuscript and ensure a more comprehensive understanding of the results.

Character Independence and Modularity:
Tests for Character Independence: It is important to consider whether the characters included in the matrix are independent or exhibit modularity. Testing for independence can help ensure that the characters used in the analysis provide a reliable phylogenetic signal. Studies, such as those by Sansom et al. (2017) and Pattinson et al. (2015), have demonstrated the importance of examining character independence and sampling across partitions to achieve reliable phylogenetic estimations. Including tests for character independence or modularity in the study would strengthen the robustness of the findings.
Detailed Description of Characters: Providing a more detailed description of the characters used in the matrix in the Methods section will help readers understand the basis of the phylogenetic trees proposed. The detailed description should include not only the types of characters but also the rationale for their selection and any tests performed to ensure their validity.

Visual Evidence for Synapomorphies:
Inclusion of Images: The identification of synapomorphies is a significant aspect of the study. Including images of these synapomorphies would provide clearer visual evidence and enhance the manuscript's impact. Visual representation of these traits would make it easier for readers to understand and verify the morphological differences that define each genus.

Discussion on Apocricetus and Hattomys:
Clarification on Genus Retention: The manuscript should provide more detailed explanations regarding the decision to retain Apocricetus plinii as a separate genus and the rationale behind not merging Apocricetus and Hattomys. A thorough discussion on the morphological and phylogenetic evidence supporting this decision would add clarity and robustness to the manuscript. This discussion should consider recent advancements in phylogenetic methodologies and how they might influence the classification of these genera.

Integration of Continuous Characters:
Use of Continuous Characters: recent advancements in Bayesian phylogenetics have enabled the integration of continuous characters, such as the size (length) of M1, into the analysis. Incorporating continuous characters can provide a more nuanced understanding of trait evolution and potentially reveal more detailed phylogenetic relationships. The manuscript would benefit from discussing why continuous characters were not included and the potential implications of their inclusion in future studies.

I hope the authors find these comments interesting trying to provide a more comprehensive and detailed understanding of the phylogenetic relationships among Neogene hamsters.

·

Basic reporting

The text is clear and in well-written English. The introduction and backround show the context of the study. The references are useful and consistent. The structure of the text itself is also clear, while the figures are useful and well executed. There is a lot of additional data which are usefull to the paper.

Experimental design

This is an original study that corresponds very well to PeerJ's expectations. The problematic is well defined, and the authors have emphasized in the text the new knowledge that this study brings. The research is rigorous and uses modern methods of analysis. The description of the method, the tools used and the additional material used make it possible to replicate the study.

Validity of the findings

All data have been provided and are robust. The conclusions drawn by the authors follow the initial problem and no extrapolation is noted.

Additional comments

The authors present an interesting work based on the analysis of fossil species of cricetine rodents in a phylogenetic context. Based on a solid matrix of dental morphological characters, they comment on the relationships between species in their temporal context, and thus propose solutions to ancient taxonomic problems.
This work provides a good basis for future studies, not only based on fossil forms, but also on molecular problematics.
I therefore recommend publication of this work in PeerJ.
I have just a few comments, which I'll list below:

Comment 1: L.318: „All species belonging to Collimys (stemming from node 35) share the exclusive synapomorphy of having an ectomesolophid on the m1 (68: 1V#9)“
I think that the list of features studied (S2.net) lacks a key morphological feature. Collimys is characterised by a flat occlusal surface. Is there a reason for omitting this feature?

Comment 2: Is there a reason for not including Collimys caucasicus (see Tesakov et al. 2017)?
Reference:
Tesakov, A.S., Titov, A.S., Simakova, A.N., Frolov, P.D., Syromyatnikova, E.V., Kurshakov, S.V., Trikhunkov, Y.I., Sotnikova, M.V., Kruskop, S.V., Zelenkov, N.V., Tesakova, E.M., & Palatov, D.M. (2017). Late Miocene (Early Turolian) vertebrate faunas and associated biotic record of the Northern Caucasus: Geology, palaeoenvironment, biochronology. Fossil Imprint 73(3-4), 383-444.

Comment 3: In the same way, why not including Pseudocollimys?

Comment 4: It would also have been interesting to include or discuss the genus Colloides which is close to Collimys (see Qiu & Li 2016). Perhaps also in this context take into account Honeymys (see Martin et al. 2020).
References:
Qiu, Z., & Li, Q. (2016). Neogene Rodents from Central Nei Mongol, China. Palaeontologia Sinica C 198(30), 1–684.
Martin, R.A., Peláez-Campomanes, P., Ronez, C., Barbière, F., Kelly, T.S., Lindsay, E.H., Baskin, J.A., Czaplewski, N.J., & Pardiñas, U.F.J. (2020). A new genus of cricetid rodent (Rodentia: Cricetidae) from the Clarendonian (Late Miocene) of North America and a consideration of Sigmodontine origins. Paludicola 12(4), 298-329.

Comment 5: L. 390: „...and a later lineage Collimys hiri-Collimys longidens-Collimys dobosi. Our results, nevertheless, do not support the lineages proposed by Prieto and Rummel (2009a,b)“
Doubts about the validity of this lineage were expressed by Prieto et al. 2014 (Page 15: ‘(1) the value of the lineage C. hiri-C. dobosi is in need of confirmation, because regional variation cannot be ruled out,’). This was echoed by Hír et al. 2017 (Page 10: However, Prieto et al. (2014b) consider that this lineage is not suitable for biostratigraphic purposes at present,basically because of conflicts with the first occurrence of Microtocricetus molassicus. Moreover, while size differences are clear between the German species, this cannot be easily extrapolated at a larger scale. Whether or not the genus is oversplitted cannot be decided at present, but the morphological similarities of the above-listed species at least indicate a relationship of the central andeastern European faunas around the transition betweenthe middle and the late Miocene).
References:
Prieto, J., Angelone, C., Casanovas-Vilar, I., Gross, M., Hír, J., Hoek Ostende, L.W.v.d., Maul, L.C., & Vasilyan, D. (2014). The small mammals from Gratkorn: an overview. Palaeobiodiversity and Palaeoenvironments 94, 135-162.
Hír, J., Venczel, M., Codrea, V., Rössner, G.E., Angelone, C., van den Hoek Ostende, L.W., Rosina, V.V., Kirscher, U., & Prieto, J. (2017). Badenian and Sarmatian s.str. from the Carpathian area: Taxonomical notes concerning the Hungarian and Romanian small vertebrates and report on the ruminants from the Felsőtárkány Basin. Comptes Rendus Palevol 16(3), 312-332.

Comment 6: S1.docx: The dating of the Hammerschmiede levels is based on the deposits that are currently visible (Hammerschmiede 4 and 5 in particular). In fact it is very difficult to extrapolate the position of Hammerschmiede 1 and 3 in this profile (see details in Kirscher et al. 2016).
Reference:
Kirscher, U., Prieto, J., Bachtadse, V., Abdul Aziz, H., Doppler, G., Hagmaier, M., & Böhme, M. (2016). A biochronologic tie-point for the base of the Tortonian stage in European terrestrial settings: Magnetostratigraphy of the topmost Upper Freshwater Molasse sediments of the North Alpine Foreland Basin in Bavaria (Germany). Newsletters on Stratigraphy 49(3), 445-467

Comment 7: In the list of characters studied (S2.net) you use protoloph rather than protolophule.

---

## Round 0.2 · accepted · Accept

I confirm that the authors addressed all the reviewers' comments and I am happy with the revised version. In my opinion, the manuscript is ready for publication.

·

Basic reporting

See previous review

Experimental design

See previous review

Validity of the findings

See previous review

Additional comments

The authors have taken into account the comments of the reviewers and justified their choices. In my opinion, their argumentation is coherent.